

# Exploring regulatory networks in plants: transcription factors of starch metabolism

Cristal López-González[1], Sheila Juárez-Colunga[1],
Norma Cecilia Morales-Elías[1] and Axel Tiessen[1,2]

[1] Departamento de Ingeniería Genética, CINVESTAV Unidad Irapuato, Irapuato, México
[2] Laboratorio Nacional PlanTECC, Irapuato, México

## ABSTRACT

Biological networks are complex (non-linear), redundant (cyclic) and compartmentalized at the subcellular level. Rational manipulation of plant metabolism may have failed due to inherent difficulties of a comprehensive understanding of regulatory loops. We first need to identify key factors controlling the regulatory loops of primary metabolism. The paradigms of plant networks are revised in order to highlight the differences between metabolic and transcriptional networks. Comparison between animal and plant transcription factors (TFs) reveal some important differences. Plant transcriptional networks function at a lower hierarchy compared to animal regulatory networks. Plant genomes contain more TFs than animal genomes, but plant proteins are smaller and have less domains as animal proteins which are often multifunctional. We briefly summarize mutant analysis and co-expression results pinpointing some TFs regulating starch enzymes in plants. Detailed information is provided about biochemical reactions, TFs and cis regulatory motifs involved in sucrose-starch metabolism, in both source and sink tissues. Examples about coordinated responses to hormones and environmental cues in different tissues and species are listed. Further advancements require combined data from single-cell transcriptomic and metabolomic approaches. Cell fractionation and subcellular inspection may provide valuable insights. We propose that shuffling of promoter elements might be a promising strategy to improve in the near future starch content, crop yield or food quality.

# INTRODUCTION

Plant cells are autotrophic organisms fully exposed to many environmental signals. While plants must cope with a wide range of conditions (e.g., light, temperature, water availability, etc.), animals enjoy more stable environments since they are able to escape from danger and to migrate searching for food. Plants are totipotent while animal cells are non-totipotent due to regulatory restrictions by cytosolic and nuclear factors. Photosynthesis in plants leads to sucrose and starch providing food for heterotrophic organisms. This review summarizes what we know about transcriptional regulation of starch metabolism in flowering plants. Most genes of starch synthesis and degradation

Corresponding author
Axel Tiessen,
atiessen@ira.cinvestav.mx

have been widely studied due to their importance for plant physiology and growth (*Zhang et al., 2012*). The expression of key enzymes and their regulatory mechanism at different levels have been investigated (*Sakulsingharoj et al., 2004*; *Li et al., 2011c*; *Gámez-Arjona et al., 2011*). However, their regulation at transcriptional level is still unclear (*Kötting et al., 2010*; *Geigenberger, 2011*). The difficulty may arise by the great number of genes (isozymes) that catalyze the main key biochemical reactions in autotrophic organisms (*Tiessen & Padilla-Chacon, 2013*; *Huang, Hennen-Bierwagen & Myers, 2014*). This review starts by listing relevant enzymes and then proceeds to clarify some paradigms of biological networks. It continues with examples of gene co-expression analysis that have pinpointed some transcription factors (TFs) in plant cells. It concludes by stating the need of more molecular information by performing single cell transcription analysis combined with metabolic profiling at the subcellular level. The systematic characterization of all TFs and cis regulatory elements of starch metabolism might provide a promising avenue for rational crop improvement.

## Survey methodology

The review started with an electronic literature survey that was expanded iteratively. Scientific articles were searched in PubMed, ISI Web of Science, Google Scholar and other databases such as EndNote and Mendeley. The first search terms included following key words: starch metabolism, TFs, regulation and plants. The abbreviated names of genes and the enzyme commission (EC) numbers of key reactions of starch metabolism were also included in the literature survey. The search also included the names of the first and senior authors of publications in high impact journals during the last 20 years about starch metabolism. The pathway of sucrose to starch conversion has been intensively investigated mainly in Arabidopsis and in potato ((*Stitt & Zeeman, 2012*) and references therein).

## Comprehensive list of starch enzymes

Starch metabolism is a network of reversible biochemical reactions that is orchestrated by more than 20 proteins annotated with an EC number as depicted in Fig. 1. For some of those enzymes there are both cytosolic and plastidial isoforms. Some cytosolic isoforms are bound to the outer plastidial membrane allowing for metabolic channeling (*Satoh et al., 2008*; *Hejazi, Steup & Fettke, 2012*; *Kunz et al., 2014*; *Fettke & Fernie, 2015*; *Malinova et al., 2017*; *Nakamura et al., 2017*). Isoform expression and sugar signaling depend on the subcellular compartment, cell type, tissue and stage of development (*Tiessen & Padilla-Chacon, 2013*).

## Starch synthesis in leaves and in storage organs

Green leaves synthesize starch inside the chloroplast using ATP and F6P provided directly by the Calvin Cycle (Fig. 1). Reproductive organs like growing tubers, seeds and fruits depend on the supply of sucrose imported via the phloem by mass flow (*Rockwell, Gersony & Holbrook, 2018*). Incoming sucrose is then used for growth, cell wall deposition, respiration and storage processes such as starch biosynthesis in the plastid.

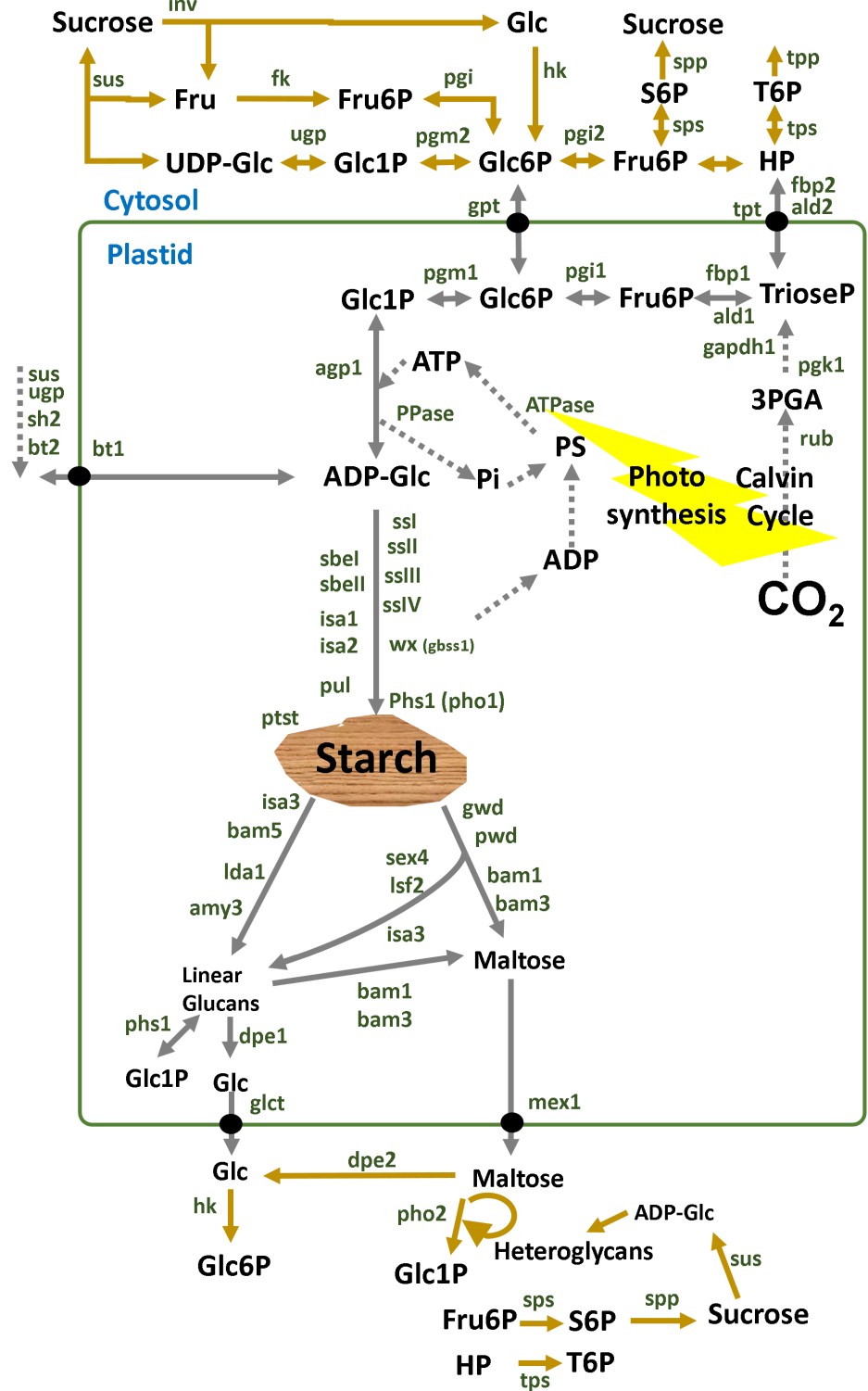

**Figure 1 Overview of starch enzymes.** Starch metabolism is a network of biochemical reactions that is orchestrated by some key enzymes such as ADP-glucose pyrophosphorylase (AGPase, EC:2.7.7.27), starch synthase (SS, EC:2.4.1.21), granule bound starch synthase (GBSS, EC:2.4.1.242), starch branching enzyme (SBE, EC:2.4.1.18), starch debranching enzyme (DBE, EC:3.2.1.196), α-amylase (AMY,

**Figure 1 (continued)**
EC:3.2.1.1), β-amylase (BAM, EC:3.2.1.2) and many other enzymes and factors (*Lloyd, Kossmann & Ritte, 2005*; *Comparot-Moss & Denyer, 2009*; *Tetlow & Emes, 2011*; *Stitt & Zeeman, 2012*). Alkaline pyrophosphatase (PPase, E.C. 3.6.1.1) catalyzes the cleavage of pyrophosphate (PPi) to orthophosphate (Pi) inside the plastid shifting the equilibrium of the AGPase reaction towards starch synthesis (*Gross & Ap-Rees, 1986*). Additional enzymes such as the alpha-glucan water dikinase (GWD, EC:2.7.9.4), the phospho-glucan water dikinase (PWD, EC:2.7.9.5), disproportionating enzyme (DPE, EC:2.4.1.25), isoamylase (ISA, EC:3.2.1.68), and α-glucan phosphorylase (PHS, EC:2.4.1.1) are also involved in the breakdown of starch (*Streb & Zeeman, 2012*). Membrane transporters participate in the metabolic network connecting several subcellular compartments such as the ATP transporter (ATT), hexose-phosphate translocator (HPT), glucose translocator (GLT) and maltose exporter (MEX1) (*Purdy et al., 2013*; *Ryoo et al., 2013*; *Stritzler et al., 2017*; *Liang et al., 2018*). Cytosolic enzymes are involved such as invertase (INV, EC:3.2.1.26), sucrose synthase (SUS, EC:2.4.1.13), hexokinase (HK, EC:2.7.1.1), fructokinase (FK, EC:2.7.1.4), glucose-6-phosphate isomerase (PGI, EC:5.3.1.9) and phosphoglucomutase (PGM, EC:5.4.2.2) (*Bahaji et al., 2015*; *Stitt & Zeeman, 2012*; *Tetlow & Emes, 2011*). In potato tubers, the adenylate-translocator imports ATP from the cytosol in counter exchange with ADP and AMP and thus provides the energy equivalents for starch synthesis (*Tjaden et al., 1998*). In sink organs, cytosolic sucrose is converted to fructose and UDP-glucose (UDPglc) through SUS in a reversible reaction (*Morell & Ap-Rees, 1986*; *Geigenberger & Stitt, 1993*; *Zrenner et al., 1995*). Using inorganic pyrophosphate (PPi) in the cytosol, fructose and UDPglc are finally processed to hexose-phosphates that can be partitioned to maintain both respiration and starch synthesis. Thereby UDP is regenerated for the SUS reaction. In potato tubers, G6P is imported to the amyloplast by an hexose phosphate translocator (HPT) (*Schott et al., 1995*; *Kammerer et al., 1998*) and converted to glucose-1-phosphate (G1P) by plastidic phosphoglucomutase (*Fernie et al., 2001*). Abreviations: Fru, fructose; Glc, glucose; Fru6P, fructose-6P; UDP-Glc, UDP-glucose; Glc1P, glucose-1P; Glc6P, glucose-6P; ADP-Glc, ADP-glucose. Enzymes are in dark green: sus1, sus2 and sus3, sucrose synthase isoform 1, 2 and 3; fk, fructokinase; pgi, glucose-6-phosphate isomerase; pgm, phosphoglucomutase; agp, ADP-glucose pyrophosphorylase; agpS, agp small subunit; agpL, agp large subunit; ssI, ssII, ssIII and ssIV, starch synthase type I, II, III and IV; pho, phosphorylase; sbeI, sbeII, starch branching enzyme I, II; isa1, isa2, isa3, isoamylase isoform 1, 2, 3; pul, pullulanase; wx (gbss1), granule bound starch synthase 1; lda1, limit dextrinase 1; amy3, alpha-amylase 3; bam1, bam2, bam3, bam5, beta-amylase isoform 1, 2, 3, 5; sex4, starch excess 4; lsf2, like sex four 2; gwd, glucan water dikinase; pwd, phosphoglucan water dikinase; phs1, plastidial starch phosphorylase 1; dpe1, dpe2, disproportionating enzyme 1, 2; glct, glucose transporter; mex1, maltose exporter.

## ADP-glucose pyrophosphorylase is a key player

ADP-glucose pyrophosphorylase is the first committed step in the starch synthesis pathway (*Smith, Denyer & Martin, 1997*). The plant enzyme is a heterotetramer, consisting of two subunits of similar size (AGPL ~51 kD, AGPS ~50 kD) (*Okita et al., 1990*). AGPase is a key enzyme exerting major control on the pathway of starch synthesis in storage as well as in photosynthetic tissue (*Tiessen et al., 2002*). The enzyme catalyzes an ATP consuming reaction, making it an exquisite candidate for regulation according to metabolic control theory (Fig. 1). Thus, the regulatory properties of this enzyme have been subject of many investigations in the past decades (*Tiessen et al., 2002, 2003*; *Kolbe et al., 2005*; *Stitt & Zeeman, 2012*). In the cereal endosperm, a cytosolic isoform of AGPase (Shrunken2 and Brittle2) and the Brittle1 transporter are the main providers of ADPglc for starch synthesis in the amyloplast (*Emes et al., 2003*; *James, Denyer & Myers, 2003*; *Tiessen et al., 2012*). Some TFs regulate the expression of several AGPase isogenes (*agpS1-2, agpL1-3*) (Table 1; Figs. 2–3).

**Table 1  Transcription factors regulating starch enzymes.**

| TF | ID | TF family | Species | Reference |
|---|---|---|---|---|
| MeERF72 | manes.15g009900 | AP2/EREB | *Manihot esculenta* | *Liu et al. (2018b)* |
| PBMY1 | Pavirv 00046166 | AP/EREB | *Panicum virgatum* | *Ambavaram et al. (2018)* |
| PBMY3 | Pavirv 00029298 | NY-F | *Panicum virgatum* | *Ambavaram et al. (2018)* |
| ZmEREB156 | GRMZM2G421033 | AP2/EREB | *Zea mays* | *Huang et al. (2016)* |
| ZmbZIP91 | GRMZM2G043600 | bZIP | *Zea mays* | *Chen et al. (2016)* |
| CRCT | LOC_Os02G15350 | bZIP | *Oryza sativa* | *Morita et al. (2015)* |
| ZmNAC36 | GRMZM2G154182 | NAC (CUC) | *Zea mays* | *Zhang et al. (2014)* |
| OsSERF1 | LOC_Os05G34730 | DREB | *Oryza sativa* | *Schmidt et al. (2014)* |
| RPBF | LOC_Os05g15350 | DOF | *Oryza sativa* | *Schmidt et al. (2014)* |
| OsbZIP58 | LOC_Os07g08420 | bZIP | *Oryza sativa* | *Wang et al. (2013)* |
| NAC96 | At5g46590 | CUC | *Arabidopsis thaliana* | *Bumee et al. (2013)* |
| WRKY75 | At5g13080 | WRKY | *Arabidopsis thaliana* | *Bumee et al. (2013)* |
| ZmDOF1 | AC155434.2_FG006 | DOF | *Zea mays* | *Noguero et al. (2013)*, *Yanagisawa (2000)* |
| ZmDOF2 | GRMZM2G009406 | DOF | *Zea mays* | *Noguero et al. (2013)*, *Yanagisawa (2000)* |
| LEC2 | At1g28300 | B3 | *Arabidopsis thaliana* | *Angeles-Núñez & Tiessen (2012)* |
| AtIDD5 | At2g02070 | IDD | *Arabidopsis thaliana* | *Ingkasuwan et al. (2012)* |
| COL | At2g21320 | COL | *Arabidopsis thaliana* | *Ingkasuwan et al. (2012)* |
| RSR1 | LOC_Os5g03040 | AP2/EREB | *Oryza sativa* | *Fu & Xue (2010)* |
| AtIDD8 | At5g44160 | IDD | *Arabidopsis thaliana* | *Seo et al. (2011)* |
| SRF1 | AB469355 | DOF | *Ipomoea batatas* | *Tanaka et al. (2009)* |
| ETR2 | AF420319 | ETR | *Oryza sativa* | *Wuriyanghan et al. (2009)* |
| OsBP-5 | | MYC-like | *Oryza sativa* | *Zhu et al. (2003)* |
| OsBP-89 | | AP2/EREB | *Oryza sativa* | *Zhu et al. (2003)* |
| SUSIBA2 | AY323206 | WRKY | *Hordeum vulgare* | *Sun (2003)* |

## Starch enzymes and plastidial proteins build metabolic complexes

Some starch biosynthetic enzymes assemble in high molecular weight complexes (*Hennen-Bierwagen et al., 2009*; *Crofts et al., 2015*). One consequence of enzyme clustering in space and time is metabolite-channeling through the formation of multienzyme assemblies known as metabolons (*Sweetlove & Fernie, 2013*). Proteins that copurified with SSIII, SSIIa, SBEIIa and SBEIIb included AGPase and SUS-SH1 forming a ~670-kD complex that may regulate carbon partitioning in developing seeds of cereals (*Hennen-Bierwagen et al., 2009*). In Arabidopsis leaves, coiled-coil proteins and PROTEIN TARGETING TO STARCH form complexes with starch synthases (SS) during granule initiation (*Seung et al., 2015*, *2017*, *2018*). Therefore, transcriptional regulation of one protein might affect the abundance of other proteins. This may be the case, for example, in the rice mutant FLOURY ENDOSPERM2 (FLO2), which pleiotropically altered the expression of many starch genes (*She et al., 2010*).

## Numerous families and multiple isoforms of starch genes

Several starch synthase isoforms use ADPglc to add its glucose moiety to amylose and amylopectin molecules in the ordered and crystalline structure of the starch granule

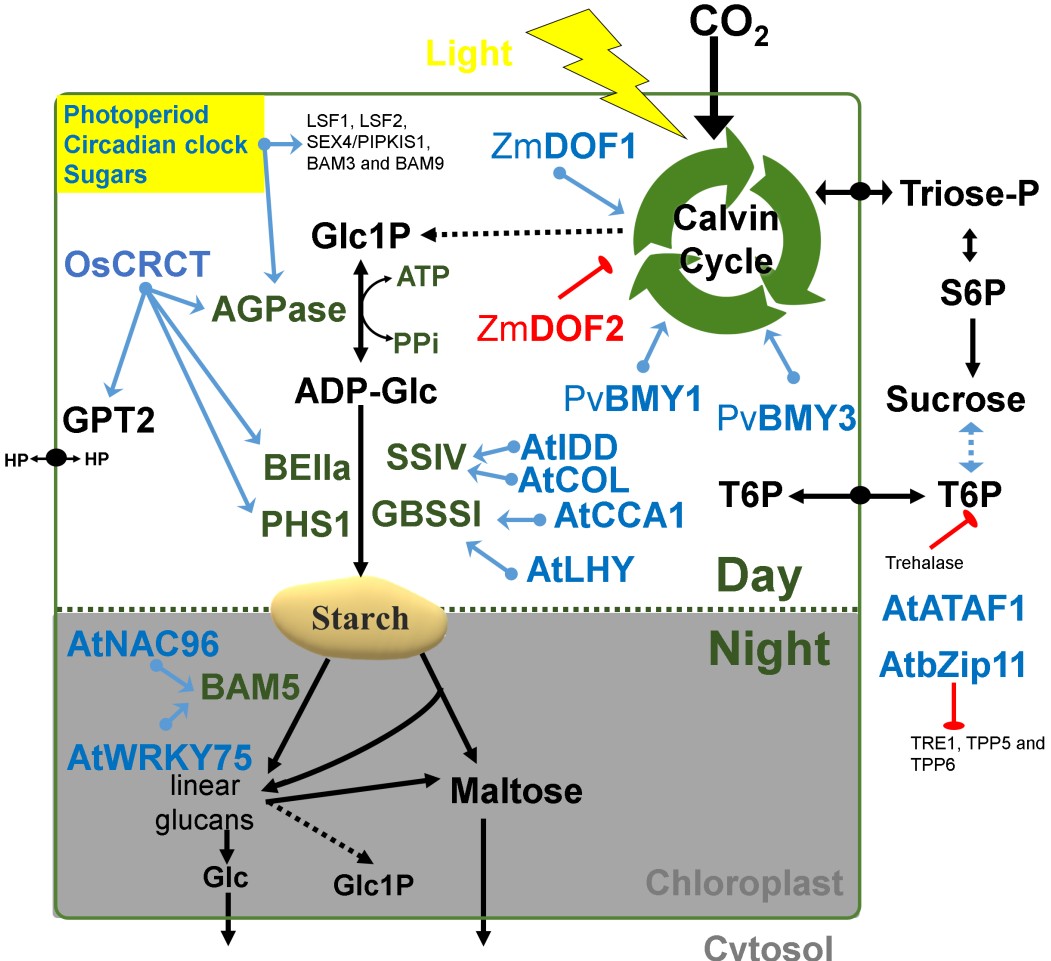

**Figure 2 Regulatory factors of starch metabolism in leaves.** Metabolites are in black letters while TFs are in blue or red color indicating activation or repression. Abbreviations: AGPase, ADP-glucose pyrophosphorylase; AtATAF1, *Arabidopsis thaliana* Transcription Activation Factor; AtCCA1, *Arabidopsis thaliana* CCA1; AtCOL, *Arabidopsis thaliana* Constant-like; AtIDD, *Arabidopsis thaliana* Indeterminate domain; AtLHY, *Arabidopsis thaliana* LATE ELONGATED HYPOCOTYL; ATP, Adenosine triphosphate; BAM, beta-amylase; BE, Branching enzyme; bZIP11, basic leucine zipper TF 11; CRCT, CO2 Responsive CCT protein; GBSS, Granule bound starch synthase; Glc, Glucose; GPT2, Glucose-phosphate translocator 2; HP, Hexose-phosphates; LSF, LIKE SEX FOUR; NAC96, NAC domain TF 96; PHS1, α-glucan phosphorylase 1; PPi, Pyrophosphate inorganic; PvBMY, *Pisum sativum* BiomassYield TF; S6P, Sucrose-6P; SEX, Starch excess; SS, Starch synthase; T6P, Trehalose-6P; TPP, Trehalose phosphatase; TRE1, Trehalase 1; WRKY75, WRKY domain TF; ZmDOF, *Zea mays* DNA binding with one finger TF.

(*Martin & Smith, 1995*; *Marshall et al., 1996*; *Smith, Denyer & Martin, 1997*; *Smith, 1999*). Different isoforms of branching enzyme and debranching enzyme are involved in the synthesis of glucans (*Ball et al., 1991*; *Zeeman et al., 1998*) (Fig. 1).

Starch synthases are divided into four subfamilies of soluble SSs (SSI, SSII, SSIII and SSIV) and one subfamily of granule-bound starch synthases (GBSS) (*Patron & Keeling, 2005*; *Leterrier et al., 2008*). Starch phosphorylase plays also an important role for starch synthesis (*Satoh et al., 2008*; *Tetlow & Emes, 2011*). Each of these enzymes are encoded by

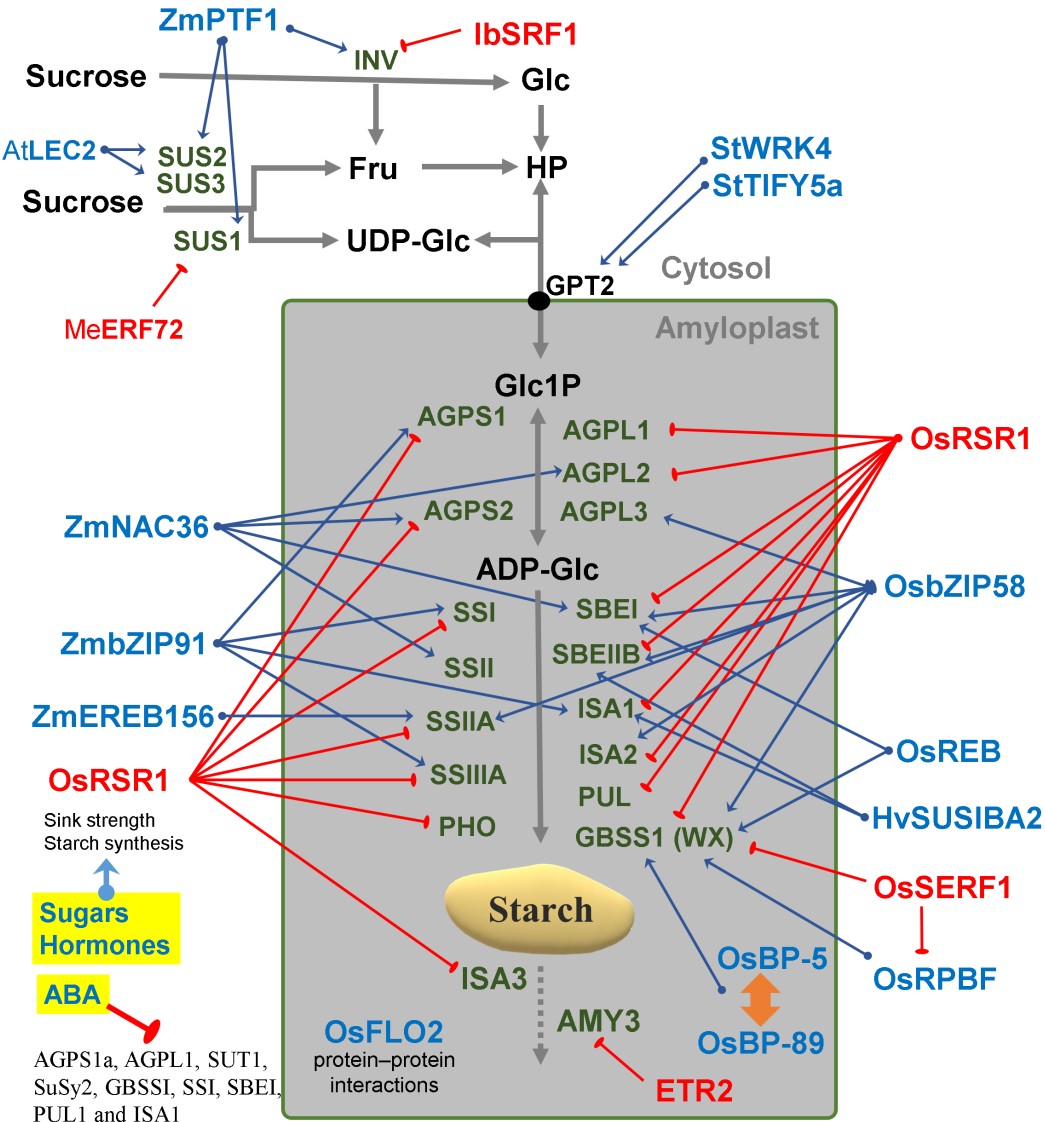

**Figure 3 Regulatory factors of starch metabolism in storage organs.** Metabolites are in black letters while TFs are in blue or red color indicating activation or repression. Abbreviations: AtLEC2, *Arabidopsis thaliana* Leafy cotyledon 2; BP-5, MYC-like TF; BP-89, Apetala2/EREB; ETR2, Subfamily II ethylene receptor; Fru, Fructose; Glc, Glucose; HP, Hexose-phosphates; HvSUSIBA2, *Hordeum vulgare* Sugar signaling in barley 2; IbSRF1, *Ipomoea batatas* Storage root factor DOF 1; MeERF72, *Manihot esculenta* Ethylene responsive factor 72; OsbZIP58, *Oryza sativa* basic leucine zipper TF 58; OsFLO2, *Oryza sativa* FLOURY ENDOSPERM2; OsRPBF, *Oryza sativa* Rice prolamin box binding factor; OsRSR1, *Oryza sativa* Rice starch regulator 1; OsSERF1, *Oryza sativa* Salt-responsive ERR1; SRF1, Storage root factor DOF TF; StTIFY5a, *Solanum tuberosum* TIFY domain 5a; StWRK4, *Solanum tuberosum* WRK4, SUS, Sucrose synthase; ZmbZIP91, *Zea mays* basic leucine zipper TF 91; ZmEREB156, *Zea mays* Ethylene response element binding protein 156; ZmNAC36, *Zea mays* NAC domain TF 36; ZmPTF1, *Zea mays* Pi starvation-induced transcription factor 1.               

many different isogenes, forming large enzyme families in plants. In maize, more than 30 genes participate in starch synthesis (*Yan et al., 2009*); while in rice are around 21 genes in total (*Hirose et al., 2006*). These isozymes have been classified by their tissue-specific expression patterns in maize and rice: type I starch genes were preferentially expressed in

endosperm (reproductive organs, sink), whereas type II starch genes were preferentially expressed in vegetative tissues (leaves, source) (*Hirose et al., 2006*; *Fu & Xue, 2010*; *Huang, Hennen-Bierwagen & Myers, 2014*).

Starch synthesis in leaves has been said to be largely similar to that in storage organs (*Santelia & Zeeman, 2011*; *Smith, 2012*; *Stitt & Zeeman, 2012*). Table 2 list some key genes in several plant species.

## Differences between metabolic and transcription networks

Metabolic and transcriptional regulation are commonly thought to be equivalent in both plant and animal systems. According to Tom Ap Rees and Mark Stitt, central metabolism of pea is like the subway map of London (*Stitt & Ap Rees, 1978*, *1980*). Certainly, compared to animal and bacterial metabolism, plant metabolism is more complex, flexible, redundant and compartmentalized (*Sweetlove & Fernie, 2013*). Even though the subcellular compartmentation of plant metabolism is thought to be well understood, unexpected results are continuously revealed by detailed gene-by-gene studies (*Lunn, 2006*). Usually, metabolic pathways are not as linear as depicted in most textbooks (*Kruger, Hill & Ratcliffe, 1999*; *Berg, Tymoczko & Stryer, 2006*). Instead of metabolic pathways, it is more accurate to speak of metabolic networks.

There are some important differences between metabolic and transcriptional networks that must be taken into account when trying to explore them by correlation analysis. Plant cells may produce a larger number of chemically distinct metabolites (~10,000) than the number of enzymes encoded by their DNA (~5,000). In metabolic networks, connections (chemical reactions) are theoretically reversible, bidirectional and may have certain stoichiometry (Fig. 4A). Metabolites can be chemically interconverted between each other, while genes are fixed entities. In transcriptional networks, some genes are more important than others; some proteins are regulatory while others are structural. Therefore, in gene networks, connections are one-directional arrows that have a certain hierarchy (Fig. 4B). From a biochemical perspective, metabolites are structurally much more diverse than genes that are all built from the same four letters (nucleotides). But from the functional and regulatory point of view, the opposite is true: Metabolites can be interconverted and are therefore more or less "equal." Genes on the contrary are "non-equal"; some have a higher hierarchy than others (Fig. 4). One transcription factor may regulate a gene coding for an enzyme but not vice versa. Many genes do the metabolic work but itself do not regulate DNA transcription or RNA translation. Thus, in transcriptional networks there are different types of genes: regulator genes and endpoint genes (Fig. 4B). Among the regulator genes, some have higher authority, since they may command many genes (both structural and regulatory genes) and are thus considered higher level factors (master switches). Connections in metabolic networks should be represented by bi-directional arrows that have a certain stoichiometry and mass action ratio but no hierarchy (Fig. 4A). In metabolic networks, in addition to standard connections (chemical reactions with an EC number), there may be regulatory connections related to allosteric regulation of enzymes, most frequently positive feed forward loops or negative feedback inhibition loops (Fig. 4A).

**Table 2 IDs of the main starch metabolic enzymes.**

| Gene name | Protein product (Enzyme) | Maize | Rice | Arabidopsis |
|---|---|---|---|---|
| SUS1 | Sucrose synthase 1 | GRMZM2G152908 | | At5g20830 |
| SUS2 | Sucrose synthase 2 | GRMZM2G318780 | | At5g49190 |
| SUS3 | Sucrose synthase 3 | | | At4g02280 |
| PGM | Phosphoglucomutase | GRMZM2G023289 | | |
| AGPL1 | ADP-glucose pyrophosphorylase large subunit 1 | GRMZM2G429899 | LOC_Os05g50380 | At5g19220 |
| AGPL2 | ADP-glucose pyrophosphorylase large subunit 2 | GRMZM2G027955 | LOC_Os01g44220 | |
| AGPL3 | ADP-glucose pyrophosphorylase large subunit 3 | GRMZM2G144002 | LOC_Os03g52460 | |
| AGPL4 | ADP-glucose pyrophosphorylase large subunit 4 | GRMZM2G391936 | LOC_Os07g13980 | |
| AGPS1 | ADP-glucose pyrophosphorylase small subunit 1 | GRMZM2G068506 | LOC_Os09g12660 | At5g48300 |
| AGPS2 | ADP-glucose pyrophosphorylase small subunit 2 | GRMZM2G163437 | LOC_Os08g25734 | |
| SSI | Starch synthase I | GRMZM2G129451 | LOC_Os06g06560 | At5g24300 |
| SSII | Starch synthase II | GRMZM2G141399 | | At3g01180 |
| SSIIa | Starch synthase IIa | GRMZM2G348551 | LOC_Os06g12450 | At2g36390 |
| SSIIb | Starch synthase IIb | GRMZM2G032628 | LOC_Os02g51070 | |
| SSIIc | Starch synthase IIc | | LOC_Os10g30156 | |
| SSIII | Starch synthase III | | | At1g11720 |
| SSIIIa | Starch synthase IIIa | GRMZM2G141399 | LOC_Os08g09230 | |
| SSIIIb | Starch synthase IIIb | | LOC_Os04g53310 | |
| SSIV | Starch synthase IV | GRMZM2G044744 | LOC_Os01g52260 | At4g18240 |
| SSIVb | Starch synthase IVb | | LOC_Os05g45720 | |
| GBSSI | Granule-bound starch synthase I | GRMZM2G024993 | LOC_Os06g04200 | |
| GBSSII | Granule-bound starch synthase II | | LOC_Os07g22930 | At1g32900 |
| BEI | Starch branching enzyme I | GRMZM2G088753 | LOC_Os06g51084 | |
| BEII | Starch branching enzyme II | GRMZM2G032628 | | At5g03650 |
| BEIIa | Starch branching enzyme IIa | | LOC_Os04g33460 | At2g36390 |
| BEIIb | Starch branching enzyme IIb | | LOC_Os02g32660 | |
| ISA1 | Starch debranching enzyme: Isoamylase I | GRMZM2G138060 | LOC_Os08g40930 | At2g39930 |
| ISA2 | Starch debranching enzyme: Isoamylase II | | LOC_Os05g32710 | At1g03310 |
| ISA3 | Starch debranching enzyme: Isoamylase III | GRMZM2G150796 | LOC_Os09g29404 | At4g09020 |
| PUL | Starch debranching enzyme: Pullulanase | GRMZM2G158043 | LOC_Os04g08270 | |
| PHOH | Starch phosphorylase H | | LOC_Os01g63270 | |
| PHOL | Starch phosphorylase L | | LOC_Os03g55090 | |
| DPE1 | Disproportionating enzyme I | | LOC_Os07g43390 | At5g64860 |
| DPE2 | Disproportionating enzyme II | | LOC_Os07g46790 | At2g40840 |
| GWD1 | Glucan water dikinase | | | At1g10760 |
| PHS1 | Plastidial starch phosphorylase 1 | | | At3g29320 |
| PHS2 | Plastidial starch phosphorylase 1 | | | At3g46970 |
| PTST2 | Protein targeting to starch 2 | | OS03G0686900 | At1g27070 |
| AMY3 | α-Amylase 3 | GRMZM2G138468 | | At1g69830 |
| BAM5 | β-Amylase 5 | GRMZM2G058310 | | At4g15210 |
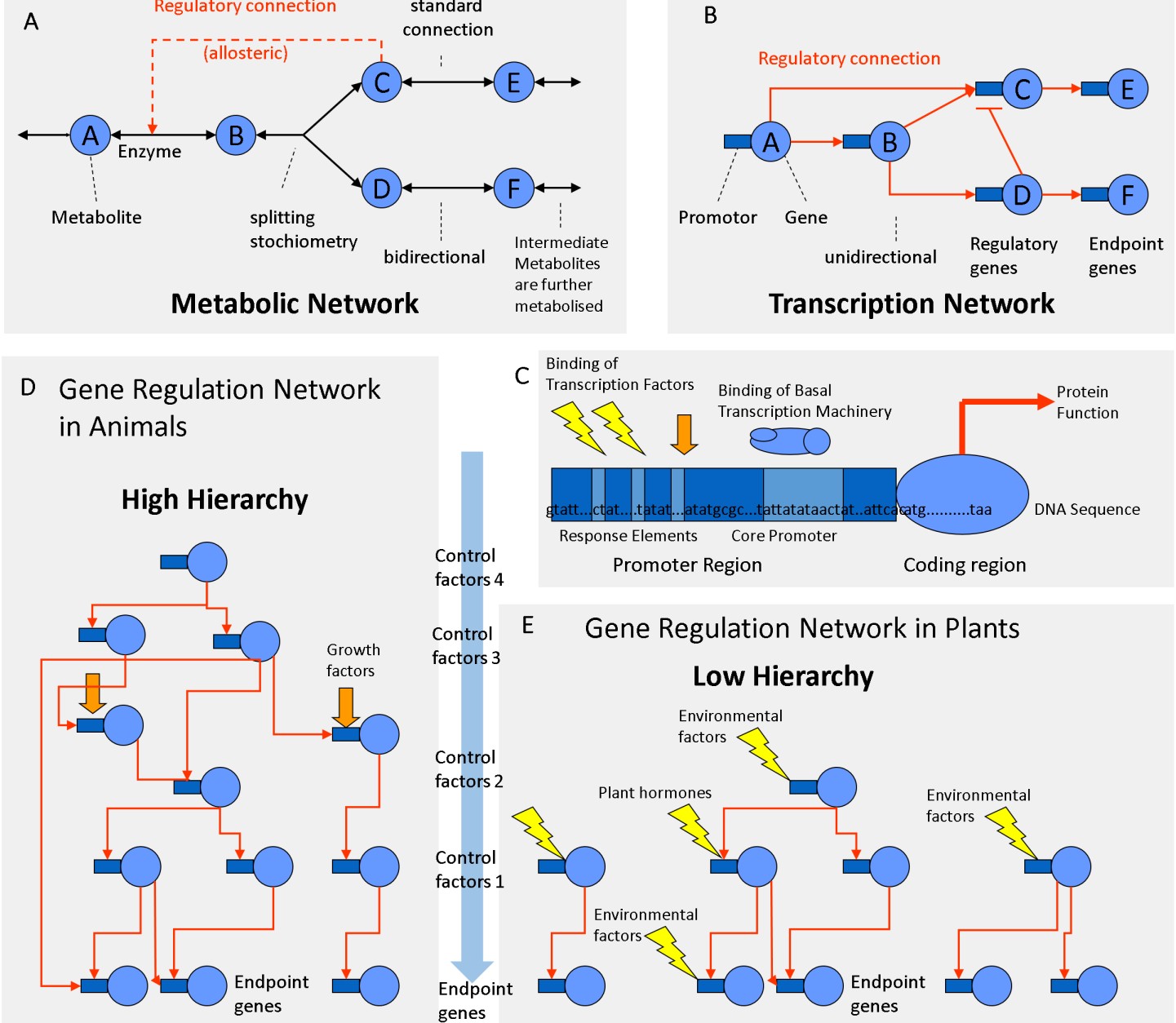

**Figure 4** **Regulation networks in plants.** (A) Metabolic network. (B) Transcriptional network. (C) Gene composed of coding determining sequence (CDS) and promoter region containing transcription factor binding elements. (D) Gene regulation network with high hierarchy (animals). (E) Gene regulation network with low hierarchy (plants).

## Differences between animal and plant protein networks

According to the classifications of gene ontology (GO) ~4–8% of the genes are involved in DNA transcription and regulation, whereas 10–20% of the genes are involved in metabolism (*Gene Ontology Consortium, 2004*; *Maere, Heymans & Kuiper, 2005*).

In plants, 5–7% of all protein-coding genes correspond to TFs (*Riaño-Pachón et al., 2007*; *Yilmaz et al., 2009*). In animal genomes, TFs make up 5–8% of the genes (*Wang & Nishida,*

2015). Plant genomes contain 34% more proteins than animal genomes (*Ramírez-Sánchez et al., 2016*). On average, an animal genome contains 25,189 proteins, whereas a plant genomes contain 36,795 proteins on average (*Ramírez-Sánchez et al., 2016*). Consequently, plant genomes code for more TFs (~1,839) than animal genomes (~1,259) (Fig. 4). The fact that plants posses more TFs is relevant for the topology of the regulatory network.

Across species there is a negative correlation between protein size and protein number in eukaryotic genomes (*Tiessen, Pérez-Rodríguez & Delaye-Arredondo, 2012*). Plant proteins are smaller and have less domains as animal proteins which are often multifunctional (*Ramírez-Sánchez et al., 2016*). Compared to the average of eukaryotic species, plants have ~34% more but ~20% smaller proteins (*Ramírez-Sánchez et al., 2016*). Compared to animal genes, plant genes have longer exons but are encoded by half the number of exons and introns (*Ramírez-Sánchez et al., 2016*). Consequently, plant proteins are simpler and have less domains and perform less complex functions (*Ramírez-Sánchez et al., 2016*). Plant transcriptional networks need to respond to a wide range of environmental inputs. Therefore, plant transcriptional networks may have more TFs that regulate gene expression with a lower hierarchy (Fig. 4E) compared to animal networks that work at a higher hierarchy (Fig. 4D). The regulatory hierarchy of plants is similar to that of one celled bacteria in that respect: flat. The consequences of the differences in the network topology can be observed at the whole organism level. Regulatory complexity becomes most evident at the tissue culture level: plant cloning can be simply done with almost any pre-differentiated vegetative cell with a mixture of auxins (roots) and cytokinin's (shoots), while regeneration and cloning of animals is harder because it requires a protected environment and a precise mixture of epigenetic, cytosolic, nuclear and membranal factors (*Zuo et al., 2017*). Co-expression analysis identified several barriers of animal cloning during somatic cell nuclear transfer (*Zuo et al., 2017*). TFs and epigenetic regulators hampered the embryo reprogramming process (*Zuo et al., 2017*). In comparison, plant cells have less barriers of transcriptional reprogramming. Therefore, plant cells are totipotent and respond to many external environmental signals, similar to bacterial cells (Fig. 4E). Animal cells are flexible and can create their own internal environment because they build tissue layers and are able to migrate between the endo-, meso- or ecto-derm in order to accommodate to better conditions. Animals make burrows, nests and liars; the blood circulatory system regulates glucose levels, oxygen, pH and temperatures in a narrow range, while plant cells are exposed to a much greater range of environmental variation. For example, desert plants adapt to diurnal variations of temperature from 5 °C in the morning to 55 °C at noon, while mammalian cells stop working if temperatures drop or rise a few degrees from 37 °C. Animals form complex organs through multiple cell layers that have a predefined cell lineage (fixed transcriptional fate). They are non-totipotent due to hierarchical restrictions by cytosolic and nuclear factors (*Zuo et al., 2017*). Animal transcription networks are more hierarchical because they react strongly to cell lineage, growth factors and cell-to-cell communication (Fig. 4D). In comparison, plant organs are less complex; the transcription networks of plant cells work less hierarchical because they respond much more directly to hormones and abiotic factors (Fig. 4E).
The number of TFs in the human genome ranges from 1,391 (*Vaquerizas et al., 2009*) to 1,639 (*Lambert et al., 2018*) while more than 2017 TFs have been reported in maize (*Burdo et al., 2014*). The Arabidopsis genome encodes >1,533 TFs, this number was 1.3 times that of *Drosophila* and 1.7 times that of *Caenorhabditis elegans* and *Saccharomyces* (*Riechmann et al., 2000*). There are many TF families that are found only in plants, such as the APETALA2/ethylene responsive element binding protein (AP2/EREBP), NAC and WRKY families; the trihelix DNA binding proteins and the auxin response factors (*Riechmann et al., 2000*). The DNA-binding with One Finger (DOF) is a group of plant-specific TFs that are implicated in stress responses, photosynthesis and flowering induction (*Noguero et al., 2013*).

## Starch transcription networks

The regulatory network involved in starch metabolism was summarized in Figs. 2–3. References of TFs and genes were listed in Tables 1 and 2. As can be seen in Figs. 2–3, the hierarchy of the regulatory network is flat, with most genes responding to hormones and environmental cues. Currently, we have limited knowledge of master TFs that with a high hierarchy regulate other TFs of starch metabolism. This contrasts with several examples of gene regulatory networks in animals that have multiple layers of hierarchical transcriptional regulation (*Cvekl & Zhang, 2017*).

The identification of TFs directly involved in the regulation of starch enzymes have been made through different strategies (mutant characterization & co-expression networks) (Tables 1 and 2). Genome-wide analysis of starch genes in potato leaves and potato tubers revealed tissue-specific expression of isoenzymes (*Van Harsselaar et al., 2017*). Therefore, we need to build regulatory schemes separately for photosynthetic and storage organs (Figs. 2 and 3).

## Transcriptional control of transitory starch in leaves

There are several interesting examples of transcriptional correlation between photosynthesis and starch biosynthesis. In maize, ZmDOF1 enhances transcription from the C4 phosphoenol pyruvate carboxylase (PEPC) promoter and ZmDOF2 blocks this transactivation and represses PEPC expression (*Yanagisawa, 2000*) (Fig. 2). In sweet potato, a DOF protein called SRF1 was found to have an indirect positive effect on starch synthesis (*Tanaka et al., 2009*) (Fig. 2). In switchgrass, PvBMY1 (BioMass Yield 1) and PvBMY3 (BioMass Yield 3) regulate photosynthesis and starch synthesis (*Ambavaram et al., 2018*). In Arabidopsis, *BAM5* is regulated by two TFs, WRKY DNA-binding domain 75 (WRKY75, At5g13080) and NAC domain-containing protein 96 (NAC096, At5g46590) (*Bumee et al., 2013*) (Fig. 2). In the *Atidd5* and *col* mutants, the reduction of *SS4* expression led to a significant increase in the number of starch granules (*Ingkasuwan et al., 2012*). In rice, CRCT was shown to positively control the expression of *BEIIa*, *OsAGPL1*, *OsAGPS1* and *GPT2*, all of which are classified as vegetative organ isoforms (*Morita et al., 2015*) (Fig. 2).

Microbial volatiles promote the accumulation of starch in leaves via a photoreceptor-mediated control (*Li et al., 2011a*). The transcriptional and post-translational regulation

network may involve NTRC-mediated changes in the redox status of plastidial enzymes (*Li et al., 2011a*).

## Transitory starch is highly responsive to the external environment

Transcripts of many starch genes are regulated by both an endogenous clock and by the diurnal cycle (i.e., light/dark cycle) (*Lu, 2005*; *Ral, 2006*) and also by sugar availability and different hormones (*Blasing et al., 2005*; *Graf & Smith, 2011*). The plant clock regulates developmental transitions like flowering, dormancy and the onset of senescence and bud break to ensure that they occur at an appropriate season or time of the day (*Flis et al., 2016*). For example, the rice GBSSII is regulated by a circadian rhythm (*Dian et al., 2003*). In Arabidopsis leaves, expression of the GBSS1 gene is controlled by two clock TFs, namely the LATE ELONGATED HYPOCOTYL (LHY) and the Myb-related CIRCADIAN CLOCK ASSOCIATED 1 (CCA1) (*Tenorio et al., 2003*) (Fig. 2).

Also, some SS isoforms are affected by photoperiods (*Lu, 2005*; *Ral, 2006*). Even though regulation of starch genes at the transcriptional level has been reported, much less is known about translational control of protein synthesis (*Kötting et al., 2010*). Diurnal changes in the transcriptome of Arabidopsis leaves revealed both transcriptional and posttranscriptional regulation of starch enzymes (*Smith, 2004*). Strong transcriptional control of starch genes occurs toward the end of the light (*Zeeman, Smith & Smith, 2007*; *Tsai et al., 2009*; *Streb & Zeeman, 2012*). Different AGPase isoforms respond differently to photoperiod, circadian clock or sugar (*Geigenberger, 2011*; *Seferoglu et al., 2013*). The Arabidopsis genes APL3 and APL4 are induced by both Suc and hexoses in leaves (*Li et al., 2002*; *Thellin et al., 2009*; *Michalska et al., 2009*). In lentil leaves, some AGPase isoforms are differentially regulated during short and long days (*Seferoglu et al., 2013*). Overall, it can be said that the expression of isogenes is certainly tissue-dependent, such as in the case of AGPase (*Huang, Hennen-Bierwagen & Myers, 2014*).

The duration of the photoperiod has two major consequences for plant growth and metabolism. Firstly, a longer night requires alterations in the timing of growth and the diurnal allocation of carbon (*Sulpice et al., 2009*, *2014*). Secondly, shorter light periods decrease growth because less light energy is available to sustain carbon fixation by photosynthesis. The transient reserves of carbon are used as a energy buffer during darkness (*Smith & Stitt, 2007*; *Stitt & Zeeman, 2012*). In Arabidopsis, expression of LSF1, LSF2, SEX4/PIPKIS1, BAM3 and BAM9 were regulated by the clock-, C- and light-signaling (*Flis et al., 2016*) (Fig. 2). At dawn, while starch biosynthesis was transcriptionally down-regulated, β-amylase was strongly up-regulated (*Flis et al., 2016*). The activity of β-amylase is associated with starch grains normally during late grain filling and also during germination (*Radchuk et al., 2017*). The rate of starch synthesis in the green leaves is increased during short photoperiods because a higher amount of carbon is required for sucrose synthesis during the long night (*Pokhilko et al., 2014*; *Sulpice et al., 2014*; *Mugford et al., 2014*). Overall, it can be said that the expression of many starch genes in photosynthetic tissues is light and time-regulated (Fig. 2), while in sink organs, transcriptional regulation might depend more upon the levels of sugars and/or phytohormones (Fig. 3).

## Plant transcription networks are highly responsive to hormones

The coordinated regulation of gene expression in sink and source sink tissues is orchestrated by light, sugars and energy status (*Geigenberger, 2011*). In addition to light and sugars, hormones and volatiles also play a key role. Ethylene and other hormones such as abscisic acid (ABA), salicylic acid and jasmonic acid are major players in coordinating signaling networks involved in the response to biotic and abiotic factors (*Foyer, Kerchev & Hancock, 2012*). The highly expressed GBSS gene was strongly repressed during ethylene-induced ripening in the banana pulp (*Zhu et al., 2011*). Also, the rice DNA-binding protein OsBP-5 forms a heterodimer with OsEBP-89, an ethylene-responsive element-binding protein that negatively regulates GBSSI expression (*Zhu et al., 2003*).

Abscisic acid treatment can promote AGPase and SS activity and decrease α-amylase and β-amylase (*Liu et al., 2018b*). ABA regulates sucrose import into the developing endosperm leading to a repression of *AGPS1a, AGPL1, SUT1, SuSy2, GBSSI, SSI, SBEI, PUL1* and *ISA1*genes (*Mukherjee et al., 2015*) (Fig. 3). An ethylene-responsive factor, ZmEREB156 is involved in the regulation of *ZmSSIIIa* in response to the synergistic effect between Suc and ABA (*Huang et al., 2016*). An ethylene receptor, ETR2, increases starch accumulation in the internodes of rice (*Wuriyanghan et al., 2009*). Overall, it can be said that the expression of many starch genes is strongly hormone- and sugar-regulated (Fig. 3).

## Transcriptional control of storage starch in tubers and seeds

In barley, SUSIBA2, a sugar-inducible TF belonging to the WRKY class, bound to the *ISA1* promoter and exhibited a similar expression pattern as *ISA1* (*Sun, 2003*) (Fig. 3). Furthermore, WRKY4 and TIFY5a (a plant-specific TF) were co-expressed with starch synthesis genes in potato tubers (*Van Harsselaar et al., 2017*) (Fig. 3). In rice it has been reported that OsSERF1 influences grain filling and starch synthesis. It binds directly to the *GBSSI* promoter and regulates *RPBF* which in turn also directly binds to *pGBSSI* (*Schmidt et al., 2014*). OsSERF1 can also negatively regulate the expression of *AGPL2, SSI, SSIIIa* and *GBSSI* (*Schmidt et al., 2014*) (Fig. 3).

## Transcriptional regulators of sucrose degradation

In sweet potato, SRF1 negatively regulates the vacuolar invertase gene (*Ibbfruct2*) (*Tanaka et al., 2009*). In cassava, MeERF72 is a negative regulator of *MeSus1* (*Liu et al., 2018a*). In Arabidopsis, *AtSUS2* and *AtSUS3* genes are down regulated by LEC2 (*Angeles-Núñez & Tiessen, 2012*). In maize, ZmPTF1 regulates *sus1, sus2, sh1B* and two invertase genes (*Li et al., 2011c*). ZmbZIP91 lowers osmotic pressure by consuming sucrose in the maize endosperm, thus increasing sucrose fixation from the source to the sink (*Chen et al., 2016*). Mutant analysis determined that FLO2 altered the expression of *SUS* and other genes of sucrose-starch metabolism in rice seeds (*She et al., 2010*). FLO2 harbors a tetratricopeptide repeat motif mediating protein–protein interactions rather than acting itself as a TF (Fig. 3).

## Co-expression networks reveal regulatory modules of starch genes

In addition to mutant studies, co-expression networks have been analyzed in Arabidopsis, rice and maize (*Tsai et al., 2009*; *Fu & Xue, 2010*; *Bumee et al., 2013*; *Chen et al., 2016*).

Genes constrained to a specific tissue and genes that are co-regulated across different samples, have been identified by simple linear correlation of transcript abundances (*Aoki, Ogata & Shibata, 2007*). Co-expression analysis is a powerful tool to identify genes, that regulate specific metabolic pathways, in a systematic manner. This analysis assumes that genes with similar expression patterns may be functionally associated (*Yonekura-Sakakibara et al., 2008*). A novel photoperiod regulatory mechanism has been coined as translational coincidence (*Seaton et al., 2018*). In maize, a co-expression network was constructed using data from 60 different stages/tissues of the inbred genotype B73. This constitutes a "developmental" network that characterizes the gene expression pattern of the organs of that crop plant. One example was the identification of ZmbZIP91 which regulates the expression of other starch genes in maize (*Chen et al., 2016*). Another example was the identification of Rice Starch Regulator 1 (RSR1) by a co-expression analysis (*Fu & Xue, 2010*). RSR1 was found to be negatively co-expressed with starch synthesis genes and was experimentally confirmed as a modulator of starch metabolic enzymes in rice (Fig. 3).

Some modules have been classified for starch biosynthesis suggesting a general transcriptional co-regulation (*Tsai et al., 2009*). Some starch genes were co-expressed with TFs of the bZIP family such as MYB, NAC (for NAM, ATAF and CUC) or AP2/EREBP families (*Fu & Xue, 2010*). In rice, a gene member of the AP2/EREBP family (RSR1) was the only one that negatively co-expressed with type I starch synthesis genes (*Fu & Xue, 2010*) (Fig. 3). In Arabidopsis, the Transcription Activation Factor1 (ATAF1) activates the expression of TREHALASE1 and leads to a sugar starvation metabolome through reduced trehalose-6-phosphate levels (Fig. 2). Coordinated transcriptional responses of starch metabolic genes triggered by ATAF1 largely overlap with expression patterns of carbon starved plants (*Garapati et al., 2015*). Starch levels were elevated in *ataf1* knockout plants and reduced in ATAF1 overexpressors (*Garapati et al., 2015*). The expression of the *TRE1*, *TPP5* and *TPP6* genes was also induced by bZIP11 (*Ma et al., 2011*) (Fig. 2).

## Cis-regulatory elements of starch metabolism

Isogenes with highly variable promoter sequences show the largest divergence in expression (*Lemmon et al., 2014*). The prominence of cis elements may indicate that cis regulation is a more effective evolutionary mechanism than *trans* regulation for adapting isogene expression to increase fitness under a changing environment (*Lemmon et al., 2014*). Therefore, a rational approach of cis element shuffling and targeted editing of promoter motifs may yield better results for crop improvement than transgenic approaches. Instead of inserting new coding determining sequences from heterologous species with strong viral promoters such as 35S, it may be safer to shuffle promoter elements and edit the untranslated regions of endogenous genes. A cisgenic fine-tuning may have less biosafety regulatory restrictions than the commercial transgenic strategy. In addition to motifs known to be present in C starvation-induced genes (CACGTG/ACGT), motifs associated with the response to hormones, sugars, light and circadian regulation are also enriched in starch genes (*Cookson et al., 2016*; *Li et al., 2018*). Bioinformatic analysis revealed regulatory cis-elements putatively responsible for the spatio-temporal pattern of *AtSUS2* expression such as the W-box (ttgact) and SEF3 (aaccca) motifs (*Angeles-Núñez & Tiessen, 2012*).

An bZIP TF called REB interacts with the ACGT elements in the promoters of both *Wx* and *SBE1* (*Cai, 2002*). A cis-acting motif with a signature of [ATC][AC][CTG][ATC]AAAGN [AC] [GCA][ATC] was found in 20 out of 24 (~83 %) of group I genes (*ISA, GWD1, SS3, GBS1, AMY3, AMY2, SBE3, ISA1, DPE2, SS2, SEX4-LIKE2, PHS1, PHS2, SEX4, BAM2, ISA3, SS4, SBE2, MEX1, SS1, GWD3, APS1, PGM1 Y DPE1*); mutation of this cis-element induced *APS1* expression in roots, indicating that this cis-element could mediate transcriptional repression (*Tsai et al., 2009*). A shifted electrophoresis band was only detected when ZmbZIP91 was incubated with the biotin-labelled ACTCAT element, which indicated that ZmbZIP91 is able to bind directly to ACTCAT elements but not TCATT elements (*Chen et al., 2016*). Some bZIP TFs (bZIP63/At5g28770, bZIP11/At4g34590, bZIP53/At3g62640, bZIP2/At2g18160 and bZIP1/At5g49450) facilitate SnRK1 signaling via their recruitment to G-box motifs (*Baena-González et al., 2007*). In rice, OsbZIP58 was shown to bind directly to the promoters of six starch-synthesizing genes, *OsAGPL3, OsWx, OsSSIIa, OsSBE1, OsBEIIb* and *OsISA2* (*Wang et al., 2013*) (Fig. 3). OsbZIP20, REB/OsbZIP33, OsbZIP34 and OsbZIP58 can bind to both the C53 and Ha-2 fragments and may regulate the expression of *SBE1* and *Wx* (*Wang et al., 2013*) (Fig. 3). In maize, ZmbZIP91 only binds to the promoters of *pAGPS1, pISA1, pSSIIIa* and *pSSI* (*Chen et al., 2016*).

## Perspectives to identify TFs related to plant yield

Identification of all TFs and cis-elements would enable a future strategy of rational metabolic design in order to turn on starch synthesis in tissues that lack starch (*Tsai et al., 2009*). Increasing crop yield has remained one of the main goals of plant breeding. The fine-tuning of CRCT expression in transgenic rice may contribute to the future development of crop varieties optimized for biorefinery purposes (*Morita et al., 2015*). In the domestication of maize from teosinte, starch metabolism in the grains was highly correlated with yield and harvest index. Many efforts have been made to increase yield by modifying the regulatory properties of key starch enzymes (*Smidansky et al., 2002, 2003; Smith, 2008; Li et al., 2011b; Kang et al., 2013*). But several first attempts have failed. In order to achieve a substantial increase in the rate of starch synthesis, the expression of a large set of enzymes and transporters need to be activated simultaneously in the pathway. This is not a simplistic one-enzyme strategy as in the first generation of transgenic plants. We need to elucidate all TFs involved in the regulation of starch metabolic enzymes. Master regulators at the post-transcriptional level have been found such as TOR1 and SNRK1 (sucrose and energy signaling). We still need to find master switches at the transcriptional level for starch metabolism. The possible existence of transcriptional "master switches" for starch is an idea not yet widely accepted among colleagues. Currently, it is assumed that starch can be synthesized whenever there is light (energy) and enough $CO_2$ inside photosynthetic leaves, or whenever enough oxygen (energy), sucrose and hormones are supplied to storage organs. However, microscopy reveals that not all cells make starch, thus we wonder why some differentiated cells are full of it while others completely lack it. With the advantage of new transcriptomic technologies, it will be possible to build regulatory networks that can help to elucidate the TFs behind

the expression patterns of starch metabolic genes. But we must solve the old problem as when studying metabolism, that whole organs and cell mixtures are homogenized and analyzed in bulk. Subcellular analysis of metabolism is needed to pinpoint key regulation sites. For example, detailed subcellular inspection using fluorescent microscopy allowed to distinguish the metabolic source of blue glow in banana leaves, fruit skin and pulp (*Tiessen, 2018*). When epidermis cells are mixed with stomatal, palisade and mesophyll cells, it will turn impossible to elucidate all TFs reliably that are responsible for the metabolic differences among those cells. Some cells have chlorophyll, sugars and starch while other not. Therefore, single cell transcriptomic data needs to be generated urgently to better understand regulation of starch metabolism in plants. Both metabolites and transcripts should be measured in the same samples always. In addition to co-expression networks, we should also take more advantage of other strategies such as yeast one hybrid and yeast two hybrid to uncover the regulatory network behind of each metabolism. Currently, there are many Arabidopsis mutant reports describing TFs altering flower development or plant morphology, whereas so much remains unknown about similar TFs regulating primary metabolism. In crop plants providing abundant food supply such as maize, there is still hope to find some master TFs controlling the energy pathway.

## CONCLUSIONS

This review highlighted the importance of distinguishing different types of biological networks, namely metabolic interconversion networks and transcriptional regulatory networks (Fig. 4). Comparisons between animal and plant transcriptional networks revealed differences in the number of genes, size of the proteins and the regulatory hierarchies. A comprehensive list of enzymes and chemical reactions that are involved in starch metabolism in plants was provided (Tables 1–2). The review focused on TFs and cis-regulatory elements that are relevant for starch synthesis and degradation. Targeted mutations of cis elements may become a breeding tool in the near future. Genetic diversity may be increased by a strategy of "rational shuffling of minimal promoter elements." Detailed information about all relevant TFs and regulatory motifs may improve plant sink strength, crop yield and food quality.

## ACKNOWLEDGEMENTS

We thank Dr. Andres Estrada-Luna for technical support in the lab and the greenhouse. We also thank Dr. Jesus Ruben Torres-Garcia, Dr. Alberto Camas-Reyes, Dr. Luz Edith Casados-Vázquez and Dr. Julio Armando Massange-Sanchez for their help.

### Funding

This work was supported by grants from the Consejo Nacional de Ciencia y Tecnologia (CONACYT Mexico) to CLG, SJC and AT. Support was also provided from the National Laboratory PlanTECC, Problemas Nacionales and Infraestructura. The authors were further supported by initial funding grants by SAGARPA through CIMMYT and the

MasAgro initiative. Funding CONACYT. PN2015-613, LN2018-293362. The funders had no role in study design, data collection and analysis, decision to publish, or preparation of the manuscript.

## Grant Disclosures

The following grant information was disclosed by the authors:
Consejo Nacional de Ciencia y Tecnologia (CONACYT Mexico).
National Laboratory PlanTECC, Problemas Nacionales and Infraestructura.
SAGARPA through CIMMYT and the MasAgro initiative.
CONACYT: PN2015-613, LN2018-293362.

## Competing Interests

Axel Tiessen is an Academic Editor for PeerJ.

## Author Contributions

- Cristal López-González conceived and designed the experiments, performed the experiments, analyzed the data, prepared figures and/or tables, authored or reviewed drafts of the paper, approved the final draft.
- Sheila Juárez-Colunga performed the experiments, approved the final draft.
- Norma Cecilia Morales-Elías performed the experiments, approved the final draft.
- Axel Tiessen conceived and designed the experiments, analyzed the data, contributed reagents/materials/analysis tools, prepared figures and/or tables, authored or reviewed drafts of the paper, approved the final draft.

## Data Availability

This is a literature review with no raw data.

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
