# Peer review of "Exploring regulatory networks in plants: transcription factors of starch metabolism"

_PeerJ, doi:10.7717/peerj.6841_

## Round 0.1 · original submission · Major Revisions

Although this is a timely review, there are too many short-comings in the present version and therefore,unfortunately, the manuscript needs major revision, shortening and restructuring. The reviewers, in particular Reviewer 1, have made useful suggestions for an improved version. I could see additional errors (Oryza and not Oriza, latin names in italics, and some grammatical mistakes. The term "Starch genes" is too simplified and should be correctly addressed etc The new version requires intensive work at all levels, including the scheme that should be divided into leaf and storage starch.

Reviewer 1 ·

Basic reporting

This review was intended to focus on the transcriptional regulation of starch metabolism. To my knowledge, this topic has not been reviewed recently, and I believe a review in this area is necessary and timely. However, this particular review has many flaws (see below) which will need to be corrected before it is considered for publication.

Overall, the article is poorly structured, lacks focus, and is unnecessarily long – making it unsuitable for the general reader. For example, the authors propose that there are "master" transcriptional switches that control starch metabolism which have not been found yet. This is an idea that is not widely accepted, and the authors could have spent more of the text discussing their perspective on why they believe such specific "switches" for starch exist and how to find them, rather than spend entire sections trying to make unhelpful generalisations about plant vs animal transcription factor networks. The comparison to regulatory networks in animal cells can be dramatically shortened (the text between lines 230-232 summarises everything said between Lines 144-224, and is sufficient).

The paragraphs detailing the influence of hormones and the discovery of TFs that regulate starch metabolism (Lines 271-344), read like lists of one-sentence summaries of each cited paper rather than building a specific argument or perspective. Moreover, each paper describes work conducted in a different species, and either in leaves or storage organs/seeds. This makes it very confusing for a general audience. Perhaps a better strategy would be to build an argument - eg: first comparing differences in starch metabolism between photosynthetic and heterotrophic tissues -> then detailing which TFs have been identified in each different tissue -> and then discussing how and why their transcriptional regulation is likely to follow different cues (eg: linked to photosynthesis or circadian clock, or by availability of incoming sucrose)? Currently, the article seems to give the reader the impression that the regulatory network is the same in all tissues (See the point about Figure 1 below), which directly contradicts the authors own conclusion that cell-specific information is necessary in the future (Lines 410-421). I would have expected any review on transcriptional regulation of starch to make clear distinctions between leaf starch, and storage starches in endosperms or tubers.

Experimental design

no comment

Validity of the findings

See above - In my opinion, a central argument or perspective is lacking, or is not delivered clearly to the reader.

Additional comments

Other major points:

Line 18: I have concerns about including a sentence like “We first need to find the master switches of primary metabolism” in the abstract. This implies that “master switches” exist, but there is no evidence to date that they do. Thus, if the authors believe such switches exist, it would be better to expand on that concept in the text rather than state this as a blunt fact in the abstract. A more acceptable alternative would be “We first need to identify key factors controlling these regulatory loops”

Line 31: I think there is something wrong with the wording here – are the authors referring to the cells or the organisms? – but in either case they are wrong. Plant cells are not sessile due to the fact that they respond to many external signals. They are sessile because they cannot move because they are bound together by cell walls. What does “create their own internal environment” mean for animal cells?? That sentence implies plant cells cannot make their own internal environments, but they do in regards to pH and ion concentrations. How totipotency is relevant in this review is also unclear. I would strongly advise the authors not to start a review with such a flawed sentence.

Line 113: “participate in the starch network” is very vague. To make it more suitable to the general reader, make it clear that in the cereal endosperm, the cytosolic AGPase IS the major ADP-Glc generating activity.

Line 117: “assembly” should be “assemble”, but also, not all starch biosynthetic enzymes assemble into HMW complexes. This generalisation should be removed. Only one example of complex formation in maize endosperm is provided, neglecting the important work by Tetlow, Emes and colleagues on complexes in other cereals.

Line 140: Since the referred papers were published, there has been vast progress in understanding new regulatory proteins involved in starch synthesis in leaves, particularly PTST proteins that form complexes with the starch synthases.

Figure 1: Firstly, this figure is far too confusing to any reader. The overlap of arrows in the central part is very difficult to read. Secondly, the figure is fundamentally flawed in that it seems to be a hybrid of the pathway of starch synthesis in heterotrophic organs with the pathway of starch synthesis and degradation pathways in leaves. Most seriously, many of the TFs indicated in this figure was shown to regulate those genes in heterotrophic tissue OR leaves, but not both! Thus it is very misleading to depict all of these things happening within a single cell. This figure needs to be split into at least two separate figures – one for the leaves and one for the heterotrophic tissues. Also: Proteins should be capitalised. AMY3 makes both linear and branched glucans. The position of SEX4 and LSF2 above the arrow implies that they directly generate linear glucans, which they don't.

Table 1: “Beta-amylase “is misspelt in the table. Check accession number for SS1 in Arabidopsis. Also, an Arabidopsis accession number is given for GBSS1, but actually, it is known that the Arabidopsis GBSS is more related to GBSS2 (Cheng et al. (2012) PLoS ONE 7(1): e30088)


Minor points:

Line 38: Zhang et al. (2012) – I can’t find this in the reference list.

Line 40: Sentences should not start with a “But”. Rather: “However, their regulation at the transcriptional…”

Line 46: “concludes” might be better than “ends”

Line 71: Is it necessary to mention “R1”? Why not just say “Additional enzymes included GWD, PWD,….” etc?

Line 108: Unclear what is meant by “strategic position”? Maybe expand and explain it is the first committed step?

Line 121: This is the first mention of PPDK. An explanation should be included in the text to make it suitable for the general reader.

Ref list: Tetlow and Emes 2011a and 2011b appear to be the same reference???

Line 164: Genes are not built from letters. They are built from base pairs.

Line 165 – omit “somehow democratic”

·

Basic reporting

No comment

Experimental design

No comment

Validity of the findings

No comment

Additional comments

The review article provides with up to date literature on the regulatory network of starch metabolism. The later is used to illustrate an original point of view of the hierarchy of plant regulatory networks in comparison with animals. The authors’ conception diverge from an anthropocentric perspective and thus contributes to the understanding of plant biology. The future directions proposed in this review (i.e. single cell transcriptomics and metabolomics, swaping of promoter elements) are particularly relevant.
The following minor points should be addressed prior to publication:
- Line 62: Please do not use the word “reversible” here. Most of the stated enzymes (e.g. starch synthases, starch branching enzymes,…) do not catalyze reversible reactions.
- Figure 1: Please replace the two-directional arrow between ADP-Glc and starch by a one-directional arrow.
- Line 117: replace “assembly” by “assemble”
- Line 161-162: “Therefore, in gene networks connections are one-directional arrows that have a certain hierarchy” should be corrected as follow: “Therefore, in gene networks, connections are one-directional arrows that have a certain hierarchy”
- Line 207-208: “In comparison, plant cell ….” should be corrected as follow: “In comparison, plant cells ….”
- Line 209: “similar as in bacterial cells” should be corrected as follow: “similar to bacterial cells”
- Line 218, 219 and 220: “TF” should be corrected as follow: “Tfs”
- Line 230: “This contrast” should be corrected as follow: “This contrasts”
- Line 243: “Also some SS isoforms are affected by photoperiods” should be corrected as follow: “Also, some SS isoforms are affected by photoperiods”
- Line 244: “…, but much less is known….” should be corrected as follow: “…, much less is known….”
- Line 252: “In lentil leaves some...” should be corrected as follow: “In lentil leaves, some...”
- Line 276: “A highly...” should be corrected as follow: “The highly...”
- Line 350: “under a changing environments” should be corrected as follow: “under a changing environment”
- Line 411: “But it” should be corrected as follow: “But if”

---

## Round 0.2 · Minor Revisions

The new version is comprehensive and much clearer. There are only a few typos as indicated in the attached version. Please check once more to eliminate similar ones.

---

## Round 0.3 · accepted · Accept

I made very few corrections and am uploaded this version for further processing while in Production.

#